# Policy Optimization with Linear Temporal Logic Constraints

**Cameron Voloshin**
Caltech

**Hoang M. Le**
Argo AI

**Swarat Chaudhuri**
UT Austin

**Yisong Yue**
Argo AI
Caltech

## Abstract

We study the problem of policy optimization (PO) with linear temporal logic (LTL) constraints. The language of LTL allows flexible description of tasks that may be unnatural to encode as a scalar cost function. We consider LTL-constrained PO as a systematic framework, decoupling task specification from policy selection, and as an alternative to the standard of cost shaping. With access to a generative model, we develop a model-based approach that enjoys a sample complexity analysis for guaranteeing both task satisfaction and cost optimality (through a reduction to a reachability problem). Empirically, our algorithm can achieve strong performance even in low-sample regimes.

## 1 Introduction

The standard reinforcement learning (RL) framework aims to find a policy that minimizes a cost function. The premise is that this scalar cost function can completely capture the task specification (known as the "reward hypothesis" [55, 53]). To date, almost all theoretical understanding of RL is focused on this cost minimization setting (e.g., [62, 32, 31, 57, 45, 9, 24, 19, 10, 3, 4, 40, 47, 48]).

However, capturing real-world task specifications using scalar costs can be challenging. For one, real-world tasks often consist of objectives that are required, as well as those that are merely desirable. By combining these objectives into scalar costs, one erases the distinction between these two categories of tasks. Also, there is recent theoretical evidence that certain tasks are simply not reducible to scalar costs [1] (see Section 2). In practice, one circumvents these challenges using heuristics such as adding "breadcrumbs" [54]. However, such heuristics can lead to catastrophic failures in which the learning agent ends up exploiting the cost function in an unanticipated way [49, 61, 28, 68, 44].

In response to these limitations, recent work has studied alternative RL paradigms that use Linear Temporal Logic (LTL) to specify tasks (see Section 7). LTL is a modeling language that can express desired characteristics of future paths of the system [11]. The notation is precise enough to allow the specification of both the required and desired behaviors; the cost minimization is left only to discriminate between which LTL-satisfying policy is "best". This ensures that the main objective — e.g., time, energy, or effort — does not have any relation to the task and is easily interpretable.

Existing work on RL with LTL constraints tends to make highly restrictive assumptions. Examples include (i) known mixing time of the optimal policy [23], (ii) the assumption that every policy satisfies the task eventually [64], or (iii) known optimal discount factor [26], all of which assist in task satisfaction verification. These assumptions have complex interactions with the environment, making them impractical if not impossible to calculate. The situation is made more complex by recent theoretical results [66, 7] that show that there are LTL tasks that are not PAC-MDP-learnable.

In this paper, we address these limitations through a novel policy optimization framework for RL under LTL constraints. Our approach relies on two assumptions that are significantly less restrictive than those in prior work and circumvent the negative results on RL-modulo-LTL: the availability

of a generative model of the environment and a lower bound on the transition probabilities in the underlying MDP. Under these assumptions, we derive a learning algorithm based on a reduction to a reachability problem. The reduction in our method can be instantiated with several planning procedures that handle unknown dynamics [12, 46]. We show that our algorithm offers strong constraint satisfaction guarantees and give a rigorous sample complexity analysis of the algorithm.

In summary, the contributions of this paper are:

1. We provide a novel approach to LTL-constrained RL that requires significantly fewer assumptions, and offers stronger guarantees, than previous work.

2. We develop several new theoretical tools for our analysis. These may be of independent interest.

3. We empirically validate using both infinite- and indefinite-horizon problems, and with composite specifications such as collecting items while avoiding enemies. We find that our method enjoys strong performance, often requiring many fewer samples than our worst-case guarantees.

## 2   Motivating Examples

We examine two examples where standard cost engineering cannot capture the task (Figure 1). We consider the undiscounted setting here. See [41, 1] for difficult examples for the discounted setting.

**Example 1 (Infinite Loop).** A robot is given the task of perpetually walking between the coffee room and the office (Figure 1 (Left)). To achieve this behavior, both the policy and cost-function must be history-dependent. These can be made Markovian through proper state-space augmentation and has been studied in hierarchical reinforcement learning or learning with options [38, 56]. Options engineering is laborious and requires expertise. Nevertheless, without the appropriate augmentation, any cost-optimal policy of a Markovian cost function will fail at the task. We will see in Section 3 that any LTL expression comes with automatic state-space augmentation, requiring no expert input.

**Example 2 (Safe Delivery).** The goal is to maximize the probability of safely sending a packet from one computer to another (Figure 1 (Right)). Policy 1 leads to a hacker sniffing packets but passing them through, and is unsafe. Policy 2 leads to a hacker stealing packets with probability $p > 0$, and is safe with probability $1 - p$, and is the policy that satisfies the task. For cost engineering, let $R$ and $S$ be the recurring costs of the received and stolen states. For the two policies, the avg. costs are $g_1 = R$ and $g_2 = pS + (1 - p)R$. Strangely, we must set $R > S$ in order for $g_2 < g_1$. Fortunately, optimizing any cost function constrained to satisfying the LTL specification does not suffer from this counter intuitive behavior as only policy 2 has any chance of satisfying the LTL expression.

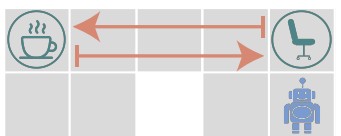 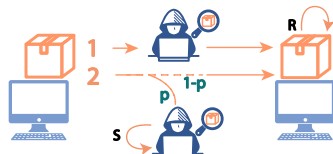

Figure 1: *(Left) Infinite Loop. The robot must perpetually walk between the coffee room and office. Without proper state-space augmentation, a markovian cost function cannot capture this task. (Right) Safe Delivery. The specification is to deliver a packet without being interfered. Policy 2 should be chosen. One would need to penalize receiving the packet significantly over having it stolen: $R > S$.*

## 3   Background and Problem Formulation

We now formulate the problem. An *atomic proposition* is a variable that takes on a truth value. An *alphabet* over a set of atomic propositions AP is given by $\Sigma = 2^{\text{AP}}$. For example, if AP $= \{a, b\}$ then $\Sigma = \{\{\}, \{a\}, \{b\}, \{a, b\}\}$. $\Delta(X)$ represents the set of probability distributions over a set $X$.

### 3.1   MDPs with Labelled State Spaces

We assume that the environment follows the finite Markov Decision Process (MDP) framework given by the tuple $\mathcal{M} = (\mathcal{S}^{\mathcal{M}}, \mathcal{A}^{\mathcal{M}}, P^{\mathcal{M}}, \mathcal{C}^{\mathcal{M}}, d_0^{\mathcal{M}}, L^{\mathcal{M}})$ consisting of a finite state space $\mathcal{S}^{\mathcal{M}}$, a finite

action space $\mathcal{A}^{\mathcal{M}}$, an ***unknown*** transition function $P^{\mathcal{M}} : \mathcal{S}^{\mathcal{M}} \times \mathcal{A}^{\mathcal{M}} \to \Delta(\mathcal{S}^{\mathcal{M}})$, a cost function $\mathcal{C} : \mathcal{S}^{\mathcal{M}} \times \mathcal{A}^{\mathcal{M}} \to \Delta([c_{\min}, c_{\max}])$, an initial state distribution $d_0 \in \Delta(\mathcal{S}^{\mathcal{M}})$, and a labelling function $L^{\mathcal{M}} : \mathcal{S}^{\mathcal{M}} \to \Sigma$. We take $\mathcal{A}^{\mathcal{M}}(s)$ to be the set of available actions in state $s$. Unlike traditional MDPs, $\mathcal{M}$ has a labeling function $L^{\mathcal{M}}$ which returns the atomic propositions that are true in that state. A **run** in $\mathcal{M}$ is a sequence of states $\tau = (s_0, s_1, \ldots)$ reached through successive transitions.

## 3.2 Linear Temporal Logic (LTL), Synchronization with MDPs, and Satisfaction

Now we give some basic background on LTL. For a more comprehensive overview, see [11].

**Definition 3.1** (LTL Specification, $\varphi$). An LTL specification $\varphi$ is the entire description of the task, including both desired and required behaviors, and is constructed from a composition of atomic propositions, including logical connectives: not ($\neg$), and ($\&$), and implies ($\to$); and temporal operators: next ($X$), repeatedly/always/globally ($G$), eventually ($F$), and until ($U$).

**Examples.** Consider again the examples in Section 2. For $AP = \{a, b\}$, some basic task specifications include safety ($G \neg a$), reachability ($Fa$), stability ($FGa$), response ($a \to Fb$), and progress ($a \,\&\, XFb$). For the Infinite Loop example (Figure 1 (Left)), $AP = \{o, c\}$ indicating the label of the grid location of our agent (office, coffee, or neither). The specification is "$GF(o \,\&\, XFc)$" meaning "go between office and coffee forever", and is a combination of safety, reachability, and progress. For the Safe Delivery example (Figure 1 (Right)), $AP = \{s\}$ indicating the safety of a state. The specification is "$Gs$" meaning "always be safe".

**LTL Satisfaction: Synchronizing MDP with LTL.** By synchronizing an MDP with an LTL formula, we can easily check if a run in the MDP satisfies a specification $\varphi$. In particular, it is possible to model the progression of satisfying $\varphi$ through a specialized automaton, an LDBA $\mathcal{B}_\varphi$ [52], defined below. More details for constructing LDBAs are in [25, 11, 35]. We drop $\varphi$ from $\mathcal{B}_\varphi$ for brevity.

**Definition 3.2.** (Limit Deterministic Büchi Automaton, LDBA [52]) An ***LDBA*** is a tuple $\mathcal{B} = (\mathcal{S}^{\mathcal{B}}, \Sigma \cup \mathcal{A}_{\mathcal{B}}, P^{\mathcal{B}}, \mathcal{S}^{\mathcal{B}*}, s_0^{\mathcal{B}})$ consisting of (i) a finite set of states $\mathcal{S}^{\mathcal{B}}$, (ii) a finite alphabet $\Sigma = 2^{\mathrm{AP}}$, $\mathcal{A}_{\mathcal{B}}$ is a set of indexed jump transitions (iii) a transition function $P^{\mathcal{B}} : \mathcal{S}^{\mathcal{B}} \times (\Sigma \cup \mathcal{A}_{\mathcal{B}}) \to 2^{\mathcal{S}^{\mathcal{B}}}$, (iv) accepting states $\mathcal{S}^{\mathcal{B}*} \subseteq \mathcal{S}^{\mathcal{B}}$, and (v) initial state $s_0^{\mathcal{B}}$. There exists a mutually exclusive partitioning of $\mathcal{S}^{\mathcal{B}} = \mathcal{S}_D^{\mathcal{B}} \cup \mathcal{S}_N^{\mathcal{B}}$ such that $\mathcal{S}^{\mathcal{B}*} \subseteq \mathcal{S}_D^{\mathcal{B}}$, and for $s \in S_D^{\mathcal{B}}, a \in \Sigma$ then $P^{\mathcal{B}}(s, a) \subseteq \mathcal{S}_D^{\mathcal{B}}$ and $|P^{\mathcal{B}}(s, a)| = 1$, deterministic. $\mathcal{A}_{\mathcal{B}}(s)$ is only (possibly) non-empty for $s \in \mathcal{S}_D^{\mathcal{B}}$ and allows $\mathcal{B}$ to transition without reading an AP. A *path* $\sigma = (s_0, s_1, \ldots)$ is a sequence of states in $\mathcal{B}$ reached through successive transitions. $\mathcal{B}$ **accepts** a path $\sigma$ if there exists some state $s \in \mathcal{S}^{\mathcal{B}*}$ in the path that is visited infinitely often.

We can now construct a synchronized product MDP from the interaction of $\mathcal{M}$ and $\mathcal{B}$.

**Definition 3.3.** (Product MDP) The product MDP $\mathcal{X}_{\mathcal{M}, \mathcal{B}} = (\mathcal{S}, \mathcal{A}, P, \mathcal{C}, d_0, L, \mathcal{S}^*)$ is an MDP with $\mathcal{S} = \mathcal{S}^{\mathcal{M}} \times \mathcal{S}^{\mathcal{B}}$, $\mathcal{A} = \mathcal{A}^{\mathcal{M}} \cup \mathcal{A}^{\mathcal{B}}$, $\mathcal{C}((m, b), a) = \mathcal{C}^{\mathcal{M}}(m, a)$ if $a \in A^{\mathcal{M}}(m)$ otherwise 0, $d_0 = \{(m, b) | m \in d_0^{\mathcal{M}}, b \in P^{\mathcal{B}}(s_0^{\mathcal{B}}, L^{\mathcal{M}}(m))\}$, $L((m, b)) = L^{\mathcal{M}}(m)$, $S^* = \{(\cdot, b) \in \mathcal{S} | b \in \mathcal{S}^{\mathcal{B}*}\}$ accepting states, and $P : \mathcal{S} \times \mathcal{A} \to \Delta(\mathcal{S})$ taking the form:

$$P((m, b), a, (m', b')) = \begin{cases} P^{\mathcal{M}}(m, a, m') & a \in A^{\mathcal{M}}(m), b' \in P^{\mathcal{B}}(b, L(m')) \\ 1, & a \in A^{\mathcal{B}}(b), b' \in P^{\mathcal{B}}(b, a), m = m' \\ 0, & \text{otherwise} \end{cases}$$

A run $\tau = (s_0, s_1, \ldots) = ((m_0, b_0), (m_1, b_1), \ldots)$ in $\mathcal{X}$ is accepting (accepted) if $(b_0, b_1, \ldots)$, the projection onto $\mathcal{B}$, is accepted. Equivalently, some $s \in \mathcal{S}^*$ in $\mathcal{X}$ is visited infinitely often. This leads us to the following definition of LTL satisfaction:

**Definition 3.4** (Satisfaction, $\tau \models \varphi$). A run $\tau$ in $\mathcal{X}$ *satisfies* $\varphi$, denoted $\tau \models \varphi$, if it is accepted.

**Definition 3.5.** (Satisfaction, $\pi \models \varphi$) A policy $\pi \in \Pi$ *satisfies* $\varphi$ with probability $\mathbb{P}[\pi \models \varphi] = \mathbb{E}_{\tau \sim \mathrm{T}_\pi^P}[\mathbf{1}_{\tau \models \varphi}]$. Here, $\mathbf{1}_X$ is an indicator variable which is 1 when $X$ is true, otherwise 0. $\mathrm{T}_\pi^P$ is the set of trajectories induced by $\pi$ in $\mathcal{X}$ with transition function $P$.

## 3.3 Problem Formulation

Our goal is to find a policy that simultaneously satisfies a given LTL specification $\varphi$ with highest probability (probability-optimal) and is also optimal w.r.t. the cost function of the MDP. We consider

(stochastic) Markovian policies $\Pi$, and define the set of all probability-optimal policies as $\Pi_{\max} = \{\arg\max_{\pi' \in \Pi} \mathbb{P}[\pi' \models \varphi]\}$. We first define the gain $g$ (average-cost) and transient cost $J$:

$$g_\pi^P \equiv \mathbb{E}_{\tau \sim \mathrm{T}_\pi^P}\left[\lim_{T \to \infty} \frac{1}{T} \sum_{t=0}^{T-1} \mathcal{C}(s_t, \pi(s_t)) \,\middle|\, \tau \models \varphi\right] , \quad J_\pi^P \equiv \mathbb{E}_{\tau \sim \mathrm{T}_\pi^P}\left[\sum_{t=0}^{\kappa_\tau} \mathcal{C}(s_t, \pi(s_t)) \,\middle|\, \tau \models \varphi\right] \quad (1)$$

where $\kappa_\tau$ is the first (hitting) time the trajectory $\tau$ leaves the transient states induced by $\pi$. When $P$ is clear from context, we abbreviate $g_\pi^P$ and $J_\pi^P$ by $g_\pi$ and $J_\pi$, respectively.

Gain optimality for infinite horizon problems has a long history in RL [12, 46]. Complementary to gain optimality, we consider a hybrid objective including the transient cost. For any $\lambda \geq 0$, the optimal policy is the probability-optimal policy with minimum combined cost:

$$\pi_\lambda^* \equiv \arg\min_{\pi \in \Pi_{\max}} J_\pi + \lambda g_\pi = \arg\min_{\pi \in \Pi_{\max}} (J_\pi + \lambda g_\pi)\mathbb{P}[\pi \models \varphi] \quad (\equiv V_{\pi,\lambda}^P). \quad \text{(OPT)}$$

In other words, probability-optimal policies are those that satisfy the entirety of the task, both desired and required behaviors, whereas $V_{\pi,\lambda}^P \equiv (J_\pi + \lambda g_\pi)\mathbb{P}[\pi \models \varphi]$ is the normalized value function[1], corresponding to a notion of energy or effort required, with $\lambda$ representing the tradeoff between gain and transient cost. We will often omit the dependence of $V$ on $P$ and $\lambda$ for brevity.

**Example.** Consider the Safe Delivery example (Figure 1 (Right)). For policy 1, $\mathbb{P}[1 \models \varphi] = 0$ and so $1 \notin \Pi_{\max}$. Let policy 2 be a cost 1 timestep before stolen or receipt, then $g_2 = R$ is the (conditional) gain, $J_2 = 1$ is the (conditional) transient costs, $\mathbb{P}[2 \models \varphi] = 1 - p$, and $V_2 = (1 + \lambda R)(1 - p)$.

**Problem 1** (Planning with Generative Model/Simulator). Suppose access to a generative model of the true dynamics $P$ from which we can sample transitions $s' \sim P(s, a)$ for any state-action pair $(s, a) \in \mathcal{S} \times \mathcal{A}$.[2] With probability $1 - \delta$, for some errors $\epsilon_\varphi, \epsilon_V > 0$, find a policy $\pi \in \Pi$ that simultaneously has the following properties: $(i)$ $|\mathbb{P}[\pi \models \varphi] - \mathbb{P}[\pi^* \models \varphi]| < \epsilon_\varphi$ $(ii)$ $|V_\pi - V_{\pi^*}| < \epsilon_V$.

# 4 Approach

## 4.1 End Components & Accepting Maximal End Components

Our analysis relies on the idea of an end component: a recurrent, inescapable set of states when restricted to a certain action set. It is a sub-MDP of a larger MDP that is probabilistically closed.

**Definition 4.1.** (End Component, EC/MEC/AMEC [11]) Consider MDP $(\mathcal{S}, \mathcal{A}, P, \mathcal{C}, d_0, L, \mathcal{S}^*)$. An end component $(E, \mathcal{A}_E)$ is a set of states $E \subseteq \mathcal{S}$ and acceptable actions $\mathcal{A}_E(s) \subseteq \mathcal{A}(s)$ (where $s \in E$) such that $\forall(s, a) \in E \times \mathcal{A}_E$ then $Post(s, a) = \{s' | P(s, a, s') > 0\} \subseteq E$. Furthermore, $(E, \mathcal{A}_E)$ is strongly connected: any two states in $E$ is reachable from one another by means of actions in $\mathcal{A}_E$. We say an end component $(E, \mathcal{A}_E)$ is *maximal* (MEC) if it is not contained within a larger end component $(E', \mathcal{A}_{E'})$, ie. $\nexists(E', \mathcal{A}_{E'})$ EC where $E \subseteq E', \mathcal{A}_E(s) \subseteq \mathcal{A}_{E'}(s)$ for each $s \in A$. A MEC $(E, \mathcal{A}_E)$ is an *accepting* MEC (AMEC) if it contains an accepting state, $\exists s \in E$ s.t. $s \in \mathcal{S}^*$.

## 4.2 High-Level Intuition

The description of our approach, LTL Constrained Planning (LCP), in Section 4.4 is rather technical in order to yield theoretical guarantees. We thus first summarize the high-level intuitions.

**Solution Decomposition.** Consider the accepting states $s_1^*, s_2^*$ in Figure 2 (Left), which are the states we need to visit infinitely often to satisfy the specification. First, let us identify the accepting maximal end components (AMECs) of $s_1^*$ and $s_2^*$: the state sets $A_1$ and $A_2$ (resp.) and their corresponding action sets $\mathcal{A}_{A_1}$ and $\mathcal{A}_{A_2}$ (the blue arrows in $A_1$ and $A_2$). Note that these AMECs do not include the yellow action in Figure 2 (Left), which has a chance of leaving $A_1$ and getting stuck in $A_3$.

Our solution first runs a *transient* policy until reaching $A_1$ or $A_2$, and then switches to a (probability-optimal) *recurrent* policy that stays within $A_1$ or $A_2$ (resp.) while visiting $s_1^*$ or $s_2^*$ (resp.) infinitely often. A probability-optimal *recurrent* policy will select actions in $\mathcal{A}_{A_1}$ and $\mathcal{A}_{A_2}$ to visit $s_1^*, s_2^*$

---

[1]Normalized objectives are not unusual in RL, e.g. in discounted settings, multiplication by $(1 - \gamma)$

[2]The use of a generative model is increasingly common in RL [24, 40, 3, 58], and is applicable in many settings where such a generative model is readily available as a simulator (e.g., [21]).

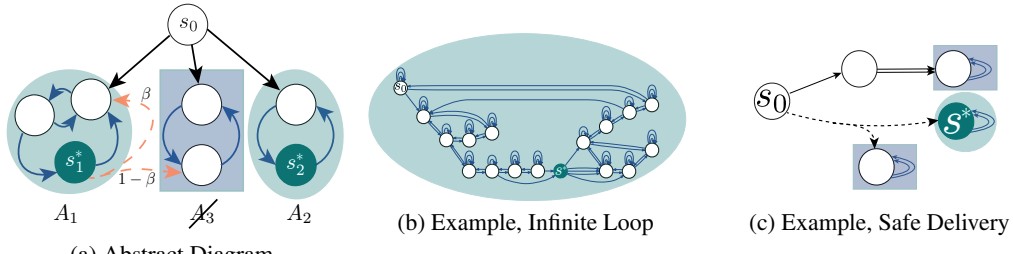

(a) Abstract Diagram    (b) Example, Infinite Loop    (c) Example, Safe Delivery

Figure 2: Product MDP diagrams. (Left) The goal of LTL Constrained Policy Optimization can be reduced to a reachability problem. We want to reach $A_1$ or $A_2$ from $s_0$ and then follow the blue arrows with some distribution. $A_3$ with the blue arrows is a rejecting end component because it does not contain an accepting state $s^*$. For $\beta < 1$, the yellow action is not in the allowable action set of $A_1$ because there is a risk of entering $A_3$, strictly decreasing our probability of LTL satisfaction. (Center) Example for Infinite Loop, Figure 1 Left. (Right) Example for Safe Delivery, Figure 1 Right.

infinitely often (e.g., the uniform policies with the AMECs $(A_1, \mathcal{A}_{A_1})$ and $(A_2, \mathcal{A}_{A_2})$). Finding a *transient* policy from $s_0$ to $A_1, A_2$ can be viewed as a reachability problem, which we can solve via a Stochastic Shortest Path (SSP) problem and leverage recent literature [58, 34].

**Cost Optimality.** As stated in OPT, the goal is to find a cost-optimal policy within the set of probability-optimal policies. For instance, the uniform policy over $\mathcal{A}_{A_1}$ and $\mathcal{A}_{A_2}$ (the blue arrows in Figure 2 (Left) is probability optimal, but may not be cost optimal. Similarly, the unconstrained cost-optimal policy may not be probability optimal. Consider just $A_1$ for the moment. Suppose the cost of the arrows between the white nodes is 4 while the other costs are 7. Then the uniform (probability-optimal) policy in $A_1$ over $\mathcal{A}_{A_1}$ has cost $\frac{1}{2}\left(\frac{4+4}{2}\right) + \frac{1}{2}\left(\frac{7+7+4}{3}\right) = 5$. The gain-optimal policy that deterministically selects the actions between the white nodes $\tilde{\pi}$ has cost $\left(\frac{4+4}{2}\right) = 4$, but is not probability optimal. If we perturb $\tilde{\pi}$ to make it even slightly stochastic (but still mostly deterministic, i.e $\eta$-greedy with $\eta \approx 0$), then it will be arbitrarily close to gain optimality and also recover probability optimality. This is a preferable probability-optimal policy over the uniform policy.

**Overall Procedure.** The high-level procedure is: (i) identify the AMECs (e.g. $(A_1, \mathcal{A}_{A_1}), (A_2, \mathcal{A}_2)$) by filtering out bad actions like the yellow arrow; (ii) find a cost-optimal (optimal gain cost) recurrent policy in each AMEC that visits some $s^*$ infinitely often; (iii) instantiate an SSP problem that finds a cost-optimal (optimal transient cost) transient policy from $s_0$ to $A_1 \cup A_2$ and avoids $A_3$; (iv) return a policy that stitches together the policies from $(ii)$ and $(iii)$. See Section 4.4 for the algorithmic details. We show in Section 5 that this solution gives the optimal solution to OPT.

### 4.3 Additional Assumptions and Definitions

Perhaps surprisingly, when planning with a simulator (i.e., generative model), even infinite data is insufficient to verify an LTL formula without having a known lower-bound on the lowest nonzero probability of the transition function $P$ [41]. Without this assumption, LTL constrained policy learning is not learnable [66]. We thus begin by assuming a known lower bound on entries in $P$.[3]

**Assumption 1** (Lower Bound). *We assume we have access to a lower bound $\beta > 0$ on the lowest non-zero probability of the transition function $P$ (Sec. 3.1):*

$$0 < \beta \leq \min_{s,a,s' \in \mathcal{S} \times \mathcal{A} \times \mathcal{S}} \{P(s,a,s') | P(s,a,s') > 0\}. \qquad (2)$$

We assume that all the costs are strictly positive, avoiding zero-cost (or negative-cost) cycles that trap a policy. Leveraging cost-perturbations and prior work [58] can remove the assumption.

**Assumption 2** (Bounds on cost function). *The minimum cost $c_{\min} > 0$ (Sec. 3.1) is strictly positive.*

Let $D = \{(s,a,s')\}$ be all the collected samples $(s,a,s')$ while running the algorithm. At any point, $\widehat{P}(s,a,s') = \frac{|\{(s,a,s') \in D\}|}{|\{(s,a) \in D\}|}$ is the empirical frequency of visiting $s'$ from $(s,a)$. We introduce

---

[3]Our assumptions are consistent with the minimal requirements studied by [41]

an event $\mathcal{E}$ and error $\psi(n)$ to quantify uncertainty on $\widehat{P}(s,a,s')$ based on current data: $n(s,a) = |\{(s,a) \in D\}|$. $\mathcal{E}$ is based on empirical Bernstein bounds [42], and holds w.p. $1 - \delta$ (Lemma B.1).

**Definition 4.2** (High Probability Event). A high probability event $\mathcal{E}$:

$$\mathcal{E} = \{\forall s, a, s' \in S \times A \times S, \forall n(s,a) > 1 : |(P(s,a,s') - \widehat{P}(s,a,s'))| \leq \psi_{sas'}(n) \leq \psi(n)\},$$

where $\psi_{sas'}(n) \equiv \sqrt{2\widehat{P}(s,a,s')(1 - \widehat{P}(s,a,s')))\xi(n)} + \frac{7}{3}\xi(n)$, $\psi(n) \equiv \sqrt{\frac{1}{2}\xi(n)} + \frac{7}{3}\xi(n)$, and $\xi(n) \equiv \log(\frac{4n^2|\mathcal{S}|^2|\mathcal{A}|}{\delta})/(n-1)$.

**Remark 4.1.** *For some $\rho > 0$, if we require $|P(s,a,s') - \widehat{P}(s,a,s')| \leq \rho$ then we need $n(s,a) = \psi^{-1}(\rho)$ samples for state-action pair $(s,a)$. See Lemma B.2 for the quantity $\psi^{-1}(\rho)$.*

**Definition 4.3** (Plausible Transition Function). The set of plausible transition functions is given by

$$\mathcal{P} = \{\tilde{P} : \mathcal{S} \times \mathcal{A} \to \Delta(\mathcal{S})| \begin{cases} \tilde{P}(s,a,s') = \widehat{P}(s,a,s'), & \widehat{P}(s,a,s') \in \{0,1\} \\ \tilde{P}(s,a,s') \in \widehat{P}(s,a,s') \pm \psi_{sas'} \cap [\beta, 1-\beta], & \text{otherwise} \end{cases} \} \quad (3)$$

Let $\mathcal{P}(s,a) \equiv \{P(s,a,\cdot)|P \in \mathcal{P}\}$ be the possible transition distributions for state-action pair $(s,a)$. We denote $P_\pi(s,s') = \mathbb{E}_{a \sim \pi}[P(s,a,s')]$ as the Markov chain given dynamics $P$ with policy $\pi$, and can be thought of as a $|\mathcal{S}| \times |\mathcal{S}|$ matrix $P_\pi = \{p_{ij}\}_{i,j=1}^{|\mathcal{S}|}$.

### 4.4 Main Algorithm: LTL Constrained Planning (LCP)

---

**Algorithm 1** LTL Constrained Planning (LCP)

---

**Param:** Error $\epsilon_V > 0$, Error $\epsilon_\varphi > 0$, Tolerance $\delta > 0$, Lower bound $\beta > 0$ (see Assumption 1)

1: Globally, track $\widehat{P}(s,a,s') = \frac{|\{(s,a,s') \in D\}|}{|\{(s,a) \in D\}|}$       // Empirical estimate of $P$
2: $((A_1, \mathcal{A}_{A_1}), \ldots, (A_m, \mathcal{A}_{A_k})) \leftarrow \texttt{FindAMEC}((\mathcal{S}, \mathcal{A}, \widehat{P}))$
3: **for** $i = 1, \ldots, k$ **do**
4:     Set $\pi_i, g_i \leftarrow \texttt{PlanRecurrent}((A_i, \mathcal{A}_{A_i}), \frac{\epsilon_V}{7\lambda})$      // Plan gain-optimal policy $\pi_i$ for $A_i$
5: Set $\pi_0 \leftarrow \texttt{PlanTransient}(((A_1, g_1), \ldots, (A_k, g_k)), \frac{2\epsilon_V}{9})$  // Plan shortest paths policy $\pi_0$ to $\cup_{i=1}^k A_i$
6: **return** $\pi = \cup_{i=0}^k \pi_i$

---

Our approach, LTL Constrained Planning (LCP), has three components, as shown in Algorithm 1 and described below. Recall from Problem 1 that the policy optimization problem OPT is instantiated over a product MDP (Def. 3.3), and that we are given a generative model of the true dynamics $P$ from which we can sample transitions $s' \sim P(s,a)$ for any state/action pair.

**Finding AMECs** (`FindAMEC`). After sampling each state-action pair $\phi_{\texttt{FindAMEC}} = O(\frac{1}{\beta})$ times (see Prop. B.4), by Assumption 1, we can verify the support of $P$. We can compute all of the MECs using Algorithm 47 from [11]. Among these MECs, we keep the AMECs, which amounts to checking if the MEC $(A_i, \mathcal{A}_{A_i})$ contains an accepting state $s^* \in \mathcal{S}^*$ from the given product MDP.

**PlanRecurrent** (PR). To plan in each AMEC $(A, \mathcal{A}_A)$ (i.e., find the optimal recurrent policy), we use Alg. 2 with (extended) relative value iteration (VI, Alg. 4 in appendix) using the optimistic Bellman operator $\mathcal{L}_{\text{PR}}^\alpha$ (see Table 1, we discuss $\alpha$ in next paragraph). Let $\pi_v$ denote the greedy policy w.r.t. the fixed point $v = \mathcal{L}_{\text{PR}}^\alpha v$ ($v$ is the optimistic value estimate). Using the $\eta$-greedy policy, $\pi \equiv (1 - \eta)\pi_v + \eta\texttt{Unif}(\mathcal{A}_A)$ (Alg. 2, Line 7), together with $P_\pi$, makes $A$ recurrent: $s^* \in A$ is visited infinitely often and $\mathbb{P}[\pi \models \varphi | s_0 \in A] = 1$. Since $\eta$ can be arbitrarily small (Lemma B.7), then $g_\pi \approx g_{\pi_v}$ and $\pi$ is both cost and probability optimal. As intuited in Section 4.2, $\pi$ has full support over $\mathcal{A}_A$ but is nearly deterministic.[4]

---

**Algorithm 2** PlanRecurrent (PR)

---

**Param:** AMEC $(A, \mathcal{A}_A)$, error $\epsilon_{\text{PR}} > 0$

1: Set $\rho \leftarrow 2\psi(\phi_{\texttt{FindAMEC}}(\beta))$  // $\rho \sim \|P - \tilde{P}\|_1^{-1}$
2: **repeat**
3:     Set $\rho \leftarrow \frac{\rho}{2}$
4:     Sample $\psi^{-1}(\rho)$ times $\forall (s,a) \in A \times \mathcal{A}_A$
5:     $v', v, \tilde{P} \leftarrow \texttt{VI}(\mathcal{L}_{\text{PR}}^\alpha, d_{\text{PR}}, \epsilon_{\text{PR}}^{\mathcal{L}})$     // $v' = \mathcal{L}_{\text{PR}}^\alpha v$
6: **until** $\rho > \frac{\epsilon_{\text{PR}}(1 - \Delta(\tilde{P}))}{3|A|c_{\max}}$  // $\|P - \tilde{P}\|_1$ *small*
7: Set policy $\pi \leftarrow \eta$-greedy policy w.r.t. $v'$
8: Set gain $g_\pi \leftarrow \frac{1}{2}(\max(v' - v) + \min(v' - v))$
9: **return** $\pi, g_\pi$

---

---
[4]Interestingly, typical RL settings admit a fully deterministic optimal policy, but for LTL constrained policy optimization the optimal policy may not be deterministic (although can be very nearly so).

Table 1: Subroutine Operators and Parameters for Value Iteration

| Op/Param | Description |
|---|---|
| $\mathcal{L}_{\text{PR}}^{\alpha} v(s)$ | $\min_{a \in \mathcal{A}_A(s)} \left( \mathcal{C}(s,a) + \alpha \min_{p \in \mathcal{P}(s,a)} p^T v \right) + (1-\alpha) v(s) \quad \forall s \in A$ |
| $d_{\text{PR}}(v_{n+1}, v_n) < \epsilon_{\text{PR}}^{\mathcal{L}}$ | $\max_{s \in A}(v_{n+1}(s) - v_n(s)) - \min_{s \in A}(v_{n+1}(s) - v_n(s)) < \frac{2\epsilon_{\text{PR}}}{3}$ |
| $\mathcal{L}_{\text{PT}} v(s)$ | $\begin{cases} \min\left\{ \min_{a \in \mathcal{A}_A(s)} \left( \mathcal{C}(s,a) + \min_{p \in \mathcal{P}(s,a)} p^T v \right), \bar{V}/\epsilon_{\varphi} \right\}, & s \in \mathcal{S} \setminus \cup_{i=1}^k A_i \\ \lambda g_i, & s \in A_i \end{cases}$ |
| $d_{\text{PT}}(v_{n+1}, v_n) < \epsilon_{\text{PT}}^{\mathcal{L}}$ | $\|v_{n+1} - v_n\|_1 < c_{\min} \epsilon_{\text{PT}} \epsilon_{\varphi}/(4\bar{V})$ |

VI in Line 5 of Alg. 2 is an iterative procedure (Alg. 4 in appendix), and terminates via $d_{\text{PR}} < \epsilon_{\text{PR}}^{\mathcal{L}}$ (Table 1). Convergence of extended VI is guaranteed [46, 29, 22], so long as the dynamics, $\tilde{P} = \arg\min_{p \in \mathcal{P}(s,a)} p^T v$, achieving the inner minimization of $\mathcal{L}_{\text{PR}}^{\alpha}$ are aperiodic – hence the aperiodicity transform $\alpha \in (0,1)$ in $\mathcal{L}_{\text{PR}}^{\alpha}$ [46]. Computing $\tilde{P}$ can be done efficiently [29] (Alg. 5 in appendix). For stability, we shift each entry of $v_n$ by the value of the first entry $v_n(0)$ [12].

Alg. 2 returns the average gain cost $g_{\pi}$ of policy $\pi$ when we have enough samples for each state-action pair in $(A, \mathcal{A}_A)$ to verify that $n > \psi^{-1}\left( \frac{\epsilon_{\text{PR}}(1-\Delta(\tilde{P}_{\pi}))}{3|A|c_{\max}} \right)$ where $\Delta(\tilde{P}_{\pi}) = \frac{1}{2} \max_{ij} \sum_k |\tilde{p}_{ik} - \tilde{p}_{jk}|$. Here, $\Delta(\tilde{P}_{\pi})$ is an easily computable measure on the ergodicity of the Markov chain $\tilde{P}_{\pi}$ [18]. We track $\psi(n)$ (recall Def. 4.2) via a variable $\rho$ and sample $\psi^{-1}(\rho) \approx \frac{1}{\rho^2}$ (see Lemma B.2) samples from each state-action pair in $(A, \mathcal{A}_A)$ (Alg. 2, Line 4). We halve $\rho$ each iteration (Alg. 2, Line 3) and convergence is guaranteed because $\rho$ will never fall below some unknown constant $\frac{\epsilon_{\text{PR}}(1-\bar{\Delta}_A)}{6|A|c_{\max}}$ (see Lemma B.8); the halving trick is required because $\bar{\Delta}_A$ is unknown a priori.

**Proposition 4.2** (PR Convergence & Correctness, Informal). *Let $\pi_A$ be the gain-optimal policy in AMEC $(A, \mathcal{A})$. Algorithm 2 terminates after at most $\log_2 \left( \frac{6|A|c_{\max}}{\epsilon_{\text{PR}}(1-\bar{\Delta}_A)} \right)$ repeats, and collects at most $n = \tilde{\mathcal{O}}(\frac{|A|^2 c_{\max}^2}{\epsilon_{\text{PR}}^2(1-\bar{\Delta}_A)^2})$ samples for each $(s,a) \in (A, \mathcal{A}_A)$. The $\eta$-greedy policy $\pi$ w.r.t. $v'$ (Alg. 2, Line 5) is gain optimal and probability optimal: $|g_{\pi} - g_{\pi_A}| < \epsilon_{PR}$, $\mathbb{P}[\pi \models \varphi | s_0 \in A] = 1$.*

---

**Algorithm 3** PlanTransient (PT)

**Param:** States & gains: $\{(A_i, g_i)\}_{i=1}^k$, err. $\epsilon_{\text{PT}} > 0$
1: Set $V_T(s) = \lambda g_i$ for $s \in A_i$    // *Terminal costs*
2: Sample $\phi_{\text{PT}}$ times $\forall (s,a) \in (\mathcal{S} \setminus \cup A_i) \times \mathcal{A}$
3: $v', v, \tilde{P} \leftarrow$ VI$(\mathcal{L}_{\text{PT}}, d_{\text{PT}}, \epsilon_{\text{PT}}^{\mathcal{L}}, V_T)$    // $v' = \mathcal{L}_{\text{PT}} v$
4: Set $\pi \leftarrow$ greedy policy w.r.t $v'$
5: **return** $\pi$

**PlanTransient (PT).** This is the stochastic shortest path (SSP) reduction step that finds a policy from the initial state $s_0$ to the AMECs (Alg. 3). The main algorithmic tool used by PlanTransient is similar to that of PlanRecurrent: it also uses extended value iteration (VI, Alg. 4 in appendix) but with a different optimistic Bellman operator $\mathcal{L}_{\text{PT}}$ (Table 1), and then returns a (fully deterministic) greedy policy w.r.t. the resulting optimistic value $v$ (Alg. 3, Line 4). $\mathcal{L}_{\text{PT}}$ is used to calculate the highest probability, lowest cost path to the AMECs (Alg. 3, Line 3).

Since rejecting end components might exist (see $A_3$ from Figure 2 (Left)), a trajectory may end up stuck and accumulate cost indefinitely, and so we must bound $\|v\|_{\infty} < \bar{V}/\epsilon_{\varphi}$ to prevent blow up. In Prop. B.13, we show how to select $\bar{V}$ such that $\pi$ will reach the target states (in this case, the AMECs), first with high prob and then with lowest cost. The existence of such a bound on $\|v\|_{\infty}$ was shown to exist, without construction, in [34]. In practice, choosing a large $\bar{V}$ is enough.

The terminal costs $V_T$ (Alg. 3, Line 1) together with Bellman equation $\mathcal{L}_{\text{PT}}$ has value function $\tilde{V}_{\pi} \approx p(J_{\pi} + \frac{1}{p} \sum_{i=1}^k p_i g_{\pi_i}) + (1-p)\bar{V}/\epsilon_{\varphi} \approx V_{\pi}$, relating to $V_{\pi}$ (OPT), see Section A.1. Here, $p_i = \mathbb{P}[\pi \text{ reaches } A_i] \equiv \mathbb{E}_{\tau \sim \text{T}_{\pi}^P}[1_{\exists s \in \tau \text{ s.t } s \in A_i}]$ and $p = \sum_{i=1}^k p_i$. VI converges when $d_{\text{PT}} < \epsilon_{\text{PT}}$ (see Table 1). Convergence of extended VI for SSP is guaranteed [58, 34]. The number of samples required for each state-action pair $(s,a) \in (\mathcal{S} \setminus \cup A_i) \times \mathcal{A}$ is $\phi_{\text{PT}} = \psi^{-1}\left( \frac{c_{\min} \epsilon_{\text{PT}} \epsilon_{\varphi}^2}{14|\mathcal{S} \setminus \cup_{i=1}^k A_i|\bar{V}^2} \right)$.

**Proposition 4.3** (PlanTransient Convergence & Correctness, Informal). *Denote the cost- and prob-optimal policy as $\pi'$. After collecting at most $n = \tilde{\mathcal{O}}(\frac{|\mathcal{S} \setminus \cup_{i=1}^k A_i|^2 \bar{V}^4}{c_{\min}^2 \epsilon_{\text{PT}}^2 \epsilon_{\varphi}^4})$ samples for each $(s,a) \in (\mathcal{S} \setminus \cup_{i=1}^k A_i) \times \mathcal{A}$, the greedy policy $\pi$ w.r.t. $v'$ (Alg. 3, Line 3) is both cost and probability optimal:*

$$\|\tilde{V}_{\pi} - \tilde{V}_{\pi'}\| < \epsilon_{PT}, \quad |\mathbb{P}[\pi \text{ reaches } \cup_{i=1}^k A_i] - \mathbb{P}[\pi' \text{ reaches } \cup_{i=1}^k A_i]| \leq \epsilon_{\varphi}.$$

## 5 End-To-End Guarantees

The number of samples necessary to guarantee an $(\epsilon_V, \epsilon_\varphi, \delta)$-PAC approximation to the cost-optimal and probability-optimal policy relies factors: $\beta$ (lower bound on the min. non-zero transition probability of $P$), $\{c_{\min}, c_{\max}\}$ (bounds on the cost function $\mathcal{C}$), $\bar{\Delta}_{A_i}$ (worst-case coefficient of ergodicity for EC $(A_i, \mathcal{A}_{A_i})$), $V$ (upper bound on the value function), and $\lambda$ (tradeoff factor).

**Theorem 5.1** (Sample Complexity). *Under the event $\mathcal{E}$, Assumption 1 and 2, after*

$$n = \tilde{\mathcal{O}}\left( \frac{1}{\beta} + \frac{1}{\epsilon_V^2}\left( \frac{|\mathcal{S}|^2 \bar{V}^4}{c_{\min}^2 \epsilon_\varphi^4} + \lambda^2 \sum_{i=1}^{k} \frac{|A_i|^2 c_{\max}^2}{(1 - \bar{\Delta}_{A_i})^2} \right) \right)$$

*samples[5] are collected from each state-action pair, the policy $\pi$ returned by Algorithm 1 is, with probability $1 - \delta$, simultaneously $\epsilon_V$-cost optimal and $\epsilon_\varphi$-probability optimal, satisfying:*

$$(i) \ \ |\mathbb{P}[\pi \models \varphi] - \mathbb{P}[\pi^* \models \varphi]| \leq \epsilon_\varphi \quad (ii) \ \ \|V_\pi - V_{\pi^*}\|_\infty < \epsilon_V. \tag{4}$$

With a sufficiently large $\lambda$ (which may not be verifiable in practice), $\pi$ is also gain optimal.

**Corollary 5.2** (Gain (Average Cost) Optimality). *There exists $\lambda^* > 0$ s.t. for $\lambda > \lambda^*$, the policy $\pi$ returned by Alg. 1 satisfies (4), $g_\pi = \arg\min_{\pi' \in \Pi_{\max}} g_{\pi'}$, and is probability and gain optimal.*

The high-level structure of our analysis follows the algorithm structure in Section 4.4, via composing the constituent guarantees. To complete the analysis, we develop some technical tools which may be of independent interest, including a gain simulation Lemma B.8 and an $\eta$-greedy optimality Lemma B.7. For ease of exposition, we also ignore paths between AMECs (see Appendix D.2).

## 6 Empirical Analysis

We perform experiments in two domains: (1) Pacman domain where an agent find food and indefinitely avoids a ghost; (2) discretized version of mountain car (MC) [14] where the agent must reach the flag. Our goal is to understand whether: (i) our LCP approach (Alg.1) produces competitive polices; (ii) LCP can work in continuous state spaces through discretization; (iii) LCP can enjoy efficient sample complexity in practice. For a baseline, we use Logically Constrained RL (LCRL, [26]), which is a Q-learning approach to LTL-constrained PO in unknown MDPs. We also do heavy cost shaping to LCRL as another baseline. See App E for more details, experiments, and figures.

### 6.1 Results

**Competitiveness of the policy in full LTL specs?** The probability of LCP satisfying the LTL spec in Figure 3 (Left) approaches 1 much faster than the two baselines. The returned policy collects the food quickly and then stays close, but avoids, the ghost. Any policy that avoids the ghost is equally good, as we have not incentivized it to stay far away. LCRL redefines cost as 1 if the LTL is solved and 0 otherwise, which is too sparse and learning suffers. Indeed, shaped LCRL performs better than straight LCRL.

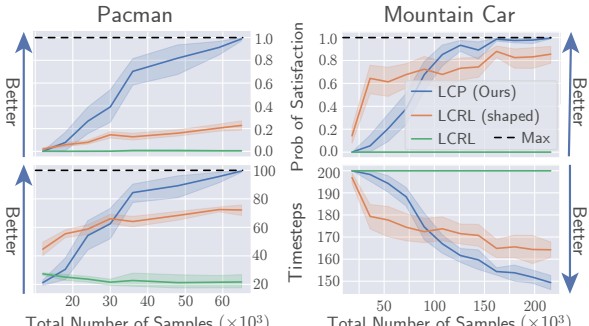

Figure 3: *Results. (Left Column) Pacman. $\varphi$ is to eventually collect food and always avoid the ghost. We let the system run for a maximum of 100 timesteps. (Right Column) Discretized Mountain Car (MC). $\varphi$ is to eventually reach the flag.*

**Performance in continuous state space?** Similarly, the probability of satisfying the LTL spec in Figure 3 (Right) goes up to 1. However, here the LCRL (shaped) baseline performs relatively well as it is being given "breadcrumbs" for how to solve the task. Our algorithm performs well without needing any cost shaping. Standard LCRL fails to learn. This experiment demonstrates that our method can be used even in discretized continuous settings.

---

[5]The lower bound relating to $\beta$ from [41] is $\Omega(\frac{\log(2\delta)}{\log(1-\beta)})$ whereas ours is $\tilde{O}(\frac{1}{\beta})$. We conjecture that $\tilde{\Omega}(\frac{1}{\beta})$ samples is required. See Appendix Section C.

**Sample Complexity?** Our theory is quite conservative w.r.t. empirical performance. In Pacman (Figure 3, Left), Thm. 5.1 suggests $\approx 350$ samples per $(s, a)$ pair just to calculate the AMECs. Empirically, LCP finds a good policy after 11 samples per $(s, a)$ pair ($\sim 66k/6k$ samples/pair).

**Other Considerations.** One of the strengths and potential drawbacks of LTL is its specificity. If a $\varphi$, for a truly infinite horizon problem, is to "eventually" do something, then accomplishing the task quickly is not required. As a finite horizon problem, in MC (Fig. 3, Right) SSP finds the fastest path to the goal. In contrast, since any stochastic policy with full support will "eventually" work, the policy returned by LCP for Fig 1 (Left) (Fig. 2 Center, & App Fig. 7) may take exponential time to complete a single loop. Two straightforward ways to address this issue are: (a) including explicit time constraints in $\varphi$; and (b) cost shaping to prefer policies reaching some $s^*$ quickly and repeatedly. Unlike standard cost-shaping, $\varphi$ satisfaction is still guaranteed since the cost is decoupled from $\varphi$.

# 7    Related Work

**Constrained Policy Optimization.** One attempt at simplifying cost functions is to split the desired behaviors from the required behaviors. The desired behaviors remain as part of the cost function while the required behaviors are treated as constraints. Recent interest in constrained policy optimization within the RL community has been related to the constrained Markov Decision Process (CMDP) framework [6, 39, 2, 43]. This framework enables clean methods and guarantees, but enforces expected constraint violations rather than absolute constraint violations. Setting and interpreting constraint thresholds can be very challenging, and inappropriate in safety-critical problems [38].

**LTL + RL.** Recently, LTL-constrained policy optimization has been developed as an alternative to CMDPs [41]. Unlike CMPDs, the entire task is encoded into an LTL expression and is treated as the constraint. Q-learning variants when dynamics are unknown and Linear Programming methods when dynamics are known are common solution concepts [50, 26, 13, 16, 20]. The Q-learning approaches rely on proper, unknowable tuning of discount factor for their guarantees. Theoretically oriented works include [23, 64]. While providing PAC-style guarantees, the assumptions made in these works rely on unknowable policy-environment interaction properties. We make no such assumptions here.

Another solution technique is employing reward machines [60, 17, 63] or high-level specifications that can be translated into reward machines [30]. These works are generally empirical and handle finite or repeated finite problems (episodic problems at test time); they can only handle a smaller set of LTL expressions, specifically regular expressions. Our work handles $\omega$-regular expressions, subsuming regular expressions and requires a nontrivial leap, algorithmically and theoretically, to access the broader set of allowable expressions. Many problems are $\omega$-regular problems, but not regular, such as liveness (something good will happen eventually) and safety (nothing bad will happen forever). The works that attempt to handle full LTL expressibility redefine reward as 1 if the LTL is solved and 0 otherwise; the cost function of the MDP is entirely ignored.

**Verification and Planning.** As an alternative to our approach, one might consider LTL satisfaction verification and extend it to an optimization technique by checking every policy (which will naively take an exponential amount of samples to verify a single policy [15, 8]). Many verification approaches exist [36, 11, 5, 67, 37, 27] and among the ones that do not assume known dynamics, the verification guarantees rely on quantities as difficult to calculate as the original verification problem itself [8].

# 8    Discussion

We have presented a novel algorithm, LCP, for policy optimization under LTL constraints in an unknown environment. We formally guarantee that the policy returned by LCP simultaneously has minimal cost with respect to the MDP cost function and maximal probability of LTL satisfaction. Our experiments verify that our policies are competitive and our sample estimates conservative.

The assumptions we make are strong, but to the best of our knowledge, are the most relaxed amongst tractable model-based algorithms proposed for this space. Model-free algorithms (Q-learning) have less stringent assumptions but do not come with the kind of guarantees that our work has and largely ignore the cost function, solving only part of the problem. An interesting future direction would be to extend our work to continuous state and action spaces and settings with function approximations.

**Acknowledgements.** Cameron Voloshin is funded partly by an NSF Graduate Fellowship and a Kortschak Fellowship. This work is also supported in part by NSF #1918865, ONR #N00014-20-1-2115, and NSF #2033851.

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
