# OpenReview forum: "Policy Optimization with Linear Temporal Logic Constraints"
_NeurIPS.cc/2022/Conference — NeurIPS 2022 Accept_

### Official Review · Reviewer_Kr29 · 2022-06-29

**Rating:** 6
**Confidence:** 3
**Soundness:** 4 excellent
**Presentation:** 2 fair
**Contribution:** 3 good

**Summary:**

The authors introduce a formal method for incorporating a probabilistic notion of LTL satisfaction for RL settings. They take the common approach to cross-product the statespace of the LTL's buchi automata for a combined reachable search space, and with the assumption of being able to sample any (s,a) pair on demand, propose a series of algorithms that separately treat the strongly connected components of the reachable states that infinitely achieve the goal condition (a requirement for infinite-trace LTL).

## Revised Score
Raised my evaluation after the authors teased apart the distinction to related work in the rebuttal.

**Questions:**

1. How might this work be modified to handle sample traces rather than arbitrary sampling of state-action pairs?
2. How might any of this work be used in settings where the reachable state-space is unrealistic to enumerate? E.g., if $|AP|>100$

**Limitations:**

The limitation of being able to arbitrarily sample from the statespace seems kind of strong. Being able to reset is a fairly strong assumption, but being able to arbitrarily sample any $(s,a)$ pair is a very strong one. I feel similarly about the number of samples required (both theoretically and empirically) across the entire state space.

There is no immediate societal impact from this work, and the authors highlight this adequately.

**Strengths And Weaknesses:**

### Originality
As far as I'm aware, the approach to uniquely treat the SCCs of the merged state space (and the strategy to move between them) is unique. The only major issue I found with the originality is the absence comparison to much of the recent work on reward machines, LTL, etc, and how they can be used to control RL. It's unclear to me if these works come with the same assumptions highlighted in the other RL+LTL works cited in the paper. Some examples include:

- Reward Machines: Exploiting Reward Function Structure in Reinforcement Learning
- LTL2Action: Generalizing LTL Instructions for Multi-Task RL
- LTL and Beyond: Formal Languages for Reward Function Specification in Reinforcement Learning
- Using Reward Machines for High-Level Task Specification and Decomposition in Reinforcement Learning


### Quality
The quality of the work appears to be of high caliber, but assessing it was difficult (more below).

### Clarity
The paper is very dense -- it felt more like an extended abstract to a JMLR paper than a well rounded and self-contained conference submission. At times, entire paragraphs are just paraphrased sentences to results described and elaborated upon in the extensive supplementary work (e.g., paragraph just before section 6). The most difficult sections include:

- Definitions 3.2 and 3.3
- Definition 4.2
- Lines 217 to 234

The general writing is extremely well done and free of grammatical / spelling issues. The only place caught my eye as a potential error is line 145 -- I think it should be $(E, A_E)$ and not $(E, A_{E'})$, and the $E \subseteq E'$ should probably be $E \subset E'$.

### Significance
While the results are strong, the significance is largely impacted by the small size of empirical domains and (ultimately) the sampling requirements for the approach to have any guarantees. It's unclear to me if these techniques can be applied in any reasonably sized RL setting (dozens to hundreds of atoms in a state).

---

> ### Author Response · Authors · 2022-08-02
> **Response to Reviewer Kr29**
>
> We thank you for your thoughtful review.
>
> **Q1:** “LCP with.. sample traces rather than arbitrary sampling…?
>
> **A1:** It can be done but would incur an exponential (non-pac) sample complexity. You can’t begin to solve the exploration-exploitation problem of RL until you are certain of the MECs: your signal for a “good” trace relies on unknowable information until you have sufficient data. Recall that you have to visit an accepting state infinitely often, this means you must be in a loop. To guarantee that you are going to make a loop forever, with high probability, takes a lot of data and requires falsification. This is all before beginning exploitation. Identification of MECs from sample traces has been briefly studied in [2014 Brazdil et al., see cited works]. We rely on arbitrary sampling to make MEC identification a simpler process.
>
> **Q2:** How might any of this work be used in settings where the reachable state-space is unrealistic to enumerate? E.g., if |AP|>100
>
> **A2:** This relation between |AP| and reachability is not too straightforward. Reachability of state-action space is related to the size of $S^M$ (MDP) and $S^B$ (Buchi Automaton) and the transition function P. The size of AP is not the most critical factor of $S^B$ but rather the “complexity” of the LTL expression. If the LTL expression is complex, this can make $S^B$ large with the exact dependence found in model-checking literature (e.g. Principles of Model Checking by Baier). That said, this method scales polynomially in $S^M * S^B$. One of the ways to scale this literature to future work is to rely on generalization from function approximation so that one doesn’t have to sample from each state-action pair.
>
> **Q3:** Reward machine literature?
>
> **A3:** In short, their work can only handle finite or repeated finite problems (episodic problems at test time). The reward machine literature handles a smaller set of LTL expressions, specifically regular expressions. Our work handles omega-regular expressions, subsuming regular expressions and requires a nontrivial leap, algorithmically and theoretically, to access the broader set of allowable expressions. Many types of problems are omega-regular problems, but not regular, such as liveness (something good will happen eventually) and safety (nothing bad will happen forever).
>
> Reward machines are useful for exposing cost function structure, such as hierarchical structure, which can increase sample efficiency through some clever off-policy tricks. The insights in these works can be applicable if our method also functioned by leveraging sample traces.

---

> > ### Comment · Reviewer_Kr29 · 2022-08-04
> > **Response to Response to Reviewer Kr29**
> >
> > Thank you for clarifying things. I think elaborating on the Reward Machine differences in the text would be useful, and please note that not all of the related work is directly leaning on reward machines. The contrast to the other works is even more vital (as it covers arbitrary LTL).
> >
> > I think the assumptions on sample complexity and form are still fairly strong, bringing down the significance, but for me this isn't a show-stopper on the work. Elaborating on these elements (and where you might remedy the issues in future work) in the camera ready would be a very useful addition.

---

> > > ### Author Response · Authors · 2022-08-08
> > > **Awknowledgement**
> > >
> > > Thank you for the response, we appreciate your feedback.
> > >
> > > Indeed, one of the works you cite is related to full LTL and not dependent on reward machines, however that work is conceptually similar to LCRL in that it redefines reward as 1 if the LTL is solved and 0 otherwise. They do some reward shaping in the same sense as we did for our LCRL baselines: there is reward for movement in the "good" (toward LTL satisfying) direction. This means that this work, and similar work, ignores the cost function of the MDP. Further, this work is largely qualitative and does not discuss performance guarantees. From a performance stand point, works relying on the redefinition of reward to be 0/1 (or some similar variant) in combination with Q learning have unverifiable requirements on the discount factor gamma for their guarantees. We thank you for pointing us to these papers and will include them in our discussion on related works.
> > >
> > > We also agree that the assumptions we make are strong, but (as far as we know) are the most relaxed amongst tractable model-based algorithms proposed for this space. Model-free algorithms (Q-learning) have less stringent assumptions but do not come with the kind of guarantees that our work has and largely ignore the cost function of the MDP (as we point out earlier), solving only part of the problem.
> > >
> > > We will make an effort to discuss the assumptions and related work with more detail in the camera ready.

---

### Official Review · Reviewer_pGRX · 2022-07-12

**Rating:** 7
**Confidence:** 3
**Soundness:** 3 good
**Presentation:** 3 good
**Contribution:** 4 excellent

**Summary:**

This paper presents a novel framework for reinforcement learning under hard LTL constraints. The proposed method, LCP, finds policies that primarily aim to satisfy the constraints over infinite horizons while secondarily minimizing a cost function (representing soft preferences). The method also relaxes many restrictive assumptions from prior works and is rigorously justified through theoretical analysis. Experiments on two RL domains help verify these claims and show the competitive performance of LCP compared to a prior method, LCRL.

**Questions:**

- Often in Markov reward processes, only the recurrent cost rate is considered since the transient costs do not affect the long-term average cost. What's the motivation for including transient costs in the optimization objective?

- (Minor issue with the framework): It's unclear to me that (OPT) is a well-defined optimization problem (with a solution $\pi^*$ that is at least as good as any other solution). This is touched on by the in-text discussion of $\eta$-greedy policies. If this is the case, then Corollary 5.2 is problematic since the result of the argmin wouldn't be defined.

- In the PacMan domain, how would incorporating the food reward into the cost, rather than the LTL constraint, affect the performance of LCP? To my understanding, either objective could specify the same desired behaviour.

**Limitations:**

The limitations were adequately addressed.

**Strengths And Weaknesses:**

**Originality**: To my understanding, this work borrows ideas from the literature (e.g. the concept of MECs in model checking is widely known) but proposes an elegant and original approach derived from these concepts. This allows the authors to derive strong theoretical guarantees, and some of the key differences with respect to prior work are explained in the text. However, I admit there may be related work that I'm unaware of.

**Quality**:
The authors propose a strong model-based approach, LCP, to the LTL-constrained policy optimization problem. This method is backed by rigorous theoretical analysis, though I was unable to verify all of the proofs (which are numerous, and quite dense). However, LCP is conceptually easy to understand and the claims appear reasonable. I have one minor issue with the framework, which I hope the authors can clarify in a question below.

The superior performance of LCP over a prior work, LCRL, are demonstrated on two domains, though I think the reasons for this need better explanation. The current claim that vanilla LCRL performs worse due to the "extremely sparse" cost function is dubious, as the cost function is uniform everywhere and virtually irrelevant in both domains. Overall, the experiments have a few other weaknesses:

- The performance of LCP appears quite sensitive to specific LTL constraints, and should be evaluated on more than only two.
- The cost function seems to play no role for LCP and vanilla LCRL in the current environments, reducing the problem to "maximize the probability of satisfying the constraint."

**Clarity**: Overall, the paper is very well-written except for the short experiments section, which delegates many important details to the Appendix. I would also appreciate if the authors could report the LBDAs and the MECs (or a brief description) for these domains so that reviewers can better understand how LCP handles these problems.

**Significance**: This paper tackles the important problem of learning constraint-satisfying policies with guarantees. The authors frame this problem in a novel setting and provide an in-depth theoretical analysis which I believe will be useful to others in the field. The claims are backed by experiments, but as of now, their significance is unclear, in part due to missing details.

---

> ### Author Response · Authors · 2022-08-02
> **Response to Reviewer pGRX**
>
> We thank you for your thoughtful review.
>
> **Q1:** The current claim that vanilla LCRL performs worse due to the "extremely sparse" cost function is dubious, as the cost function is uniform everywhere and virtually irrelevant in both domains…..reducing the problem to "maximize the probability of satisfying the constraint." … More than 2 environments?
>
> **A1:** This is a very subtle point. LCRL ignores the MDP cost function and, essentially, redefines reward as 1 if the LTL is solved and 0 otherwise. This is incredibly sparse. When we discuss “cost-shaping” of LCRL, what we mean is to incentivize movement in the Buchi Automaton so that LCRL receives some feedback in the “good” (toward LTL satisfaction) direction, as was done by the LCRL authors in their source-code (see Appendix E.1). However, cost shaping LCRL in this way, in some cases, completely invalidates the performance guarantees of LCRL. To keep our experiments fair to the baseline, we make the problems “maximize probability” problems.
>
> Further, the value function of LCRL and the value of LCP are not comparable. One is a discounted sum and the other is a, roughly, total-cost. So even if the cost function were different, the only way to compare them would be qualitatively. It is simple to construct examples where we could modify the cost function slightly and make the LCP policy prefer certain parts of the state-action space while LCP would remain insensitive.
>
> We have a few more experiments in the Appendix E.3 with further discussion on, for example, sensitivity of LCP to LTL specification within the same environment.
>
>
> **Q2:** Often in Markov reward processes, only the recurrent cost rate is considered since the transient costs do not affect the long-term average cost. What's the motivation for including transient costs in the optimization objective?
>
> **A2:** This is a great question. While modern RL undiscounted algorithms tend to focus on average cost, more sensitive cost objectives have a long rich history (see Puterman on Bias Optimality) and are largely ignored because they require complex solutions. However, reachability properties (expressible in LTL) offer a simple illustration of why transient costs are necessary to control. Suppose the LTL constraint is to reach some goal, then ideally the optimization problem should minimize the cost it takes to reach that goal. This is precisely the transient cost. Once you reach the goal, you could accumulate some additional cost to stabilize at the goal -- this is the average cost.
>
>
> **Q3:** (OPT) is a well-defined optimization problem?
>
> **A3:** This is a good point. The eta-approximation is actually to recover probability-optimality, not cost optimality. With regards to the corollary, a deterministic policy does achieve the minimum gain, but it fails to (possibly) induce an infinite loop with an accepting state. In this sense, the min over $\Pi_{\max}$ may not be achieved. We acknowledge that it may be case that we need to be a little more careful in general and possibly carry around  infs instead of mins when discussing some of our terms.  That said the point still remains: there is always a policy that is arbitrarily close to the best (possibly unattainable) one and using our method you’ll get epsilon close to that.
>
>
> **Q4:** In the PacMan domain, how would incorporating the food reward into the cost, rather than the LTL constraint, affect the performance of LCP? To my understanding, either objective could specify the same desired behavior.
>
> **A4:** One of the issues with doing this is that it requires expertise on how much reward to give for the food. By putting the requirement of “get food” in the LTL, you can drastically simplify the cost function of the MDP to be something more intuitive like amount of electricity/money or some other measure of effort. This makes the value function also more intuitive – it’s a measure of total effort.

---

> > ### Comment · Reviewer_pGRX · 2022-08-07
> > **Thanks for the response**
> >
> > Thank you for clarifying these points. I believe this is a solid contribution and I've raised my score to a 7 (though I maintain only a mid-level of confidence).

---

> > > ### Author Response · Authors · 2022-08-08
> > > **Awknowledgement**
> > >
> > > Thank you for the response, we appreciate your feedback.

---

### Official Review · Reviewer_efrT · 2022-07-15

**Rating:** 7
**Confidence:** 2
**Soundness:** 3 good
**Presentation:** 3 good
**Contribution:** 3 good

**Summary:**

Policy optimization in RL often involves complex reward engineering that can lead to unexpected results. Linear temporal logic (LTL), however, provides a robust language to encode a cost function where a scalar cost function may be unnatural. This paper provides a novel approach to LTL-constrained reinforcement learning, LTL constrained planning or LCP, under fewer assumptions and providing stronger guarantees than in the existing literature. The authors propose LCP based on a reduction to a reachability problem, and they show that the algorithm "offers strong constraint satisfaction guarantees", as well as prove its sample complexity. The algorithm itself consists of 3 parts: (1) finding an AMEC given the product MDP; (2) finding a cost-optimal recurrent policy within the AMECs; (3) finding a cost-optimal transient policy to reach this AMEC. Finally, the authors provide an empirical evaluation of their method on various environments, showing that their method achieves higher performance with lower sample complexity compared to baselines.

**Questions:**

I believe the paper, which is quite theoretical, could be improved with some discussion for practitioners, i.e. when to use LCP and when LCP would not offer extra benefit (apart from in truly infinite-horizon problems). In particular, the authors only present empirical evaluations in which LCP outperforms existing methods. Were there environments tested where LCP did not outperform existing baselines? LCP tends to give a larger advantage over baselines under which conditions (e.g. when the task is more complex)?

**Limitations:**

Limitation are discussed in Section 6, where authors describe how LCP may fall short when used alone in infinite-horizon problems.

**Strengths And Weaknesses:**

Originality & Quality: To the best of my knowledge, the method LCP appears to be new, and well-situated in / compared to the existing literature. This work is complete, proposing a method and evaluating it against existing baselines.

Clarity: The submission is clearly written and well-organized. In particular, the inclusion of a section detailing the high-level intuition is very useful to orient the reader, as the work introduces a lot of terminology and is easy to get bogged down in e.g. the proofs in the appendices.  Ideally, I would have liked to see a conclusion that recaps the takeaways from this work and that lays down an avenue for future work, but perhaps this is just a matter of style.

Significance: This work provides a unique theoretical and experimental approach to policy optimization by reframing LTL constrained policy optimization as a reachability problem to recurrent states.

---

> ### Author Response · Authors · 2022-08-02
> **Response to Reviewer efrT**
>
> We thank you for your thoughtful review.
>
> **Q1:** When to use LCP vs when no extra benefit? When did LCP not outperform existing baselines?
>
> **A1:** This work tackles full LTL expressibility in an undiscounted setting with strong performance guarantees. If either the performance guarantees or total cost (undiscounted) are not necessary to control, it is significantly easier to code up Q learning. We found that LCP outperformed the baselines in all of our experiments. LCP, as is, will not work well in conditions when a simulator and a lower bound on the min non-zero transition probability (Assumption 1) of the environment is unknown or when one needs to learn from traces.
>
> **Q2:** Conclusion?
>
> **A2:** We thank the reviewer for this suggestion. We omitted it due to space constraints and are happy to include one in the camera-ready (which has an additional page).  Briefly, some interesting discussion points include: 1) further exploring the conditions when one can establish learnability (e.g., whether Assumptions 1 & 2 are necessary); 2) whether our upper bounds on sample complexity have matching lower bounds, or whether there’s a more efficient algorithm; 3) further exploring the space of temporal logic specifications that can be optimized well using this framework (which is a more empirically focused project); and 4) studying how well our framework composes with modern learning approaches such as function approximation.

---

### Meta-Review · Area_Chair_qT61 · 2022-08-26

**Recommendation:** Accept
**Confidence:** Less certain

**Metareview:**

We have three reviews with high scores but not high confidence (confidence 2,3,3).  However, the reviews by pGRX and Kr29 seems fairly thorough with strong author responses.  I was particularly satisfied with the author responses as were the these two reviewers.  I have a lingering concern about whether the baselines for the Pacman and Mountain Car are strong enough.  But the conceptual and theoretical contribution seems to warrant publication.

**Award:**

No

---

### Decision · Program_Chairs · 2022-09-14

Accept